# Spectral Transformations and Associated Linear Functionals of the First Kind

**Juan Carlos García-Ardila** [1] and **Francisco Marcellán** [2,*]

1 Departamento de Matemática Aplicada a la Ingeniería Industrial, Universidad Politécnica de Madrid, Calle José Gutierrez Abascal 2, 28006 Madrid, Spain; juancarlos.garciaa@upm.es
2 Departamento de Matemáticas, Universidad Carlos III de Madrid, Avenida Universidad 30, 28911 Leganés, Spain
* Correspondence: pacomarc@ing.uc3m.es

**Abstract:** Given a quasi-definite linear functional $\mathbf{u}$ in the linear space of polynomials with complex coefficients, let us consider the corresponding sequence of monic orthogonal polynomials (SMOP in short) $(P_n)_{n\geq 0}$. For a canonical Christoffel transformation $\widetilde{\mathbf{u}} = (x - c)\mathbf{u}$ with SMOP $(\widetilde{P}_n)_{n\geq 0}$, we are interested to study the relation between $\widetilde{\mathbf{u}}$ and $\widetilde{\mathbf{u}^{(1)}}$, where $\mathbf{u}^{(1)}$ is the linear functional for the associated orthogonal polynomials of the first kind $(P_n^{(1)})_{n\geq 0}$, and $\widetilde{\mathbf{u}^{(1)}} = (x - c)\mathbf{u}^{(1)}$ is its Christoffel transformation. This problem is also studied for canonical Geronimus transformations.

**Keywords:** spectral transformation; Darboux transformations; first kind orthogonal polynomials; Laguerre-Hahn linear functional

**MSC:** 42C05; 15A23





## 1. Introduction and Preliminaries

The theory of orthogonal polynomials constitutes a basic topic in the framework of special functions and approximation theory. From the classical monograph by Szegő, with an analytic approach to properties of those orthogonal polynomials as the location of their zeros or their asymptotic behavior, the theory has raised great interest (see References [1–3] as excellent monographs) due to the variety of scientific fields where it constitutes a useful tool (from Numerical Analysis to Fourier series, from Mathematical Physics to Probability theory, from Coding theory to Discrete Mathematics, among others), as well as by the different approaches from the point of view of spectral problems for differential and difference operators (see Reference [4]), where classical orthogonal polynomials are studied according to their hypergeometric character and the statement of a hierarchy (Askey tableau) allowing a description of the different levels as limit cases from a level to the lower one. Another approach is based on the algebraic analysis of linear functionals and the corresponding sequences of orthogonal polynomials constituting a structural approach where the Stieltjes function associated with the linear functional play a central role. When they satisfy first order linear differential equations with polynomial coefficients, a new hierarchy (semiclassical linear functionals) is stated and you obtain a classification following a different characterization according to the degree of the polynomial coefficients involved in the above differential equation (see References [5,6]). The description of each class level constitutes an interesting problem completely solved in the lowest levels of the hierarchy (see References [6,7]). The ways to generate a level from the lowest one involves perturbations of the linear functionals which are not related to the hypergeometric approach. Indeed, the multiplication by a polynomial, the division by a polynomial, the addition of Dirac delta linear functionals and their derivatives, the symmetrization and cubic decomposition of orthogonal polynomials, the truncation of linear polynomials, among others, appear in a natural way. On the other hand, beyond the

semiclassical families, Laguerre-Hahn linear functionals (see Reference [8]) are introduced assuming the Stieltjes function satisfies a Riccati equation with polynomial coefficients. They constitute a wide class where a new hierarchy is stated following the consideration of the concept of class related, as in the semiclassical case, to the degree of the polynomial coefficients involved in the Riccati equation. Again, you go from a level to the upper one by using perturbations of linear functionals described as above, as well as new ones related to associated and anti-associated transformations. A complete classification of the lowest level has been done in Reference [8]. Examples of Laguerre-Hahn in other levels have been analyzed in References [9–12], among others.

Taking into account the sequences of orthogonal polynomials with respect to a linear functional satisfy a three-term recurrence relation, you can represent the multiplication operator in terms of such polynomial basis by using a tridiagonal matrix (known in the literature as Jacobi matrix). The eigenvalues of their principal leading submatrices are the zeros of the orthogonal polynomials; thus, you can deal with quadrature rules where the Christoffel constants are related to the norms of the eigenvectors associated with the zeros. Jacobi matrices play a central role in the analysis of perturbations of linear functionals and their consequences in quadrature formulas (see Reference [13]). In particular, LU, UL, and QR factorizations of Jacobi matrices lead to new Jacobi matrices associated with spectral linear transformation of linear functionals which show the interplay between the theory of orthogonal polynomials and matrix analysis (see References [14–16]).

In the present contribution, we deal with perturbations of linear functionals and their relevance in the Jacobi matrices and Stieltjes functions associated with them by considering two kind of transformations: the so-called Darboux transformations $T_c$ (in particular, Christoffel and Geronimus transformations) and first kind associated transformations, denoted by $T^{(1)}$. An intertwining problem between such transformations will by analyzed as the main novelty of our contribution. Thus, we present an approach involving Darboux and first associated transformations in order to generate families of rational spectral transformations which are new in the literature. Indeed, the previous contributions (References [14–16]) focus the attention on individual Darboux transformations and their relation with factorizations of the corresponding Jacobi matrices, as well as the connection between the sequences of orthogonal polynomials with respect to such linear functionals. In our work, we deal with an unified approach based on the three basic elements (Stieltjes functions, orthogonal polynomials, and Jacobi matrices) taking into account they are intimately related. As an application, such transformations are considered for Laguerre-Hahn linear functionals in order to prove that the new linear functional is again a Laguerre-Hahn linear functional, and we discuss its class. Our approach is based on the analysis of the Stieltjes function that seems to be more natural that one recently done in Reference [10], only for the Christoffel transformation, by using the functional equation that the Laguerre-Hahn linear functional satisfies.

First of all, we will provide a basic background in order to have a self-contained presentation.

Let **u** be a complex-valued linear functional defined on the linear space of polynomials with complex coefficients $\mathbb{P}$, i.e., $\mathbf{u} : \mathbb{P} \to \mathbb{C}$, $p(x) \to \langle \mathbf{u}, p(x) \rangle$. The linear functional **u** is said to be quasi-definite (respectively, positive definite) if every leading principal submatrix of the Hankel matrix $H = (\mathbf{u}_{i+j})_{i,j=0}^{\infty}$ is nonsingular (respectively, positive-definite) where, by definition, $\mathbf{u}_k =: \left\langle \mathbf{u}, x^k \right\rangle, k \in \mathbb{N}$. In this case, there exists a sequence of monic polynomials $(P_n)_{n \geq 0}$ such that $\deg P_n = n$ and $\langle \mathbf{u}, P_n(x) P_m(x) \rangle = K_n \delta_{n,m}$, where $\delta_{n,m}$ is the Kronecker symbol, and $K_n \neq 0$ (see Reference [2]). The sequence $(P_n)_{n \geq 0}$ is said to be the sequence of monic orthogonal polynomials (SMOP) with respect to **u**.

**Definition 1.** *Let* **u** *be a linear functional and* $q(x)$ *a polynomial. Then, the linear functionals* $q(x)\mathbf{u}$ *and* $(x - c)^{-m}\mathbf{u}$ *are defined, respectively, as*

$$\langle q(x)\mathbf{u}, p(x) \rangle = \langle \mathbf{u}, p(x)q(x) \rangle, \quad p \in \mathbb{P},$$

*and*

$$\left\langle (x-c)^{-m}\mathbf{u}, p(x) \right\rangle = \left\langle \mathbf{u}, \frac{p(x) - \sum\limits_{k=0}^{m-1} \frac{D^k p(c)}{k!}(x-c)^k}{(x-c)^m} \right\rangle, \quad p \in \mathbb{P},$$

*where D denotes the usual derivative operator.*

The derivative $\mathbf{u}'$ of a linear functional $\mathbf{u}$ is the linear functional defined as

$$\left\langle \mathbf{u}', p(x) \right\rangle = -\left\langle \mathbf{u}, p'(x) \right\rangle.$$

Sometimes, we also use the notation $D\mathbf{u}$ to denote the derivative of $\mathbf{u}$.

**Definition 2.** *Given the linear map $\theta_c : \mathbb{P} \to \mathbb{P}$ defined by $\theta_c p(x) = \dfrac{p(x) - p(c)}{x - c}$, we introduce the linear functional*

$$\left\langle (x-c)^{-1}\mathbf{u}, p(x) \right\rangle = \langle \mathbf{u}, \theta_c p(x) \rangle.$$

From the above Definition, we get

$$\left\langle (x-c)^{-m}\mathbf{u}, p(x) \right\rangle = \langle \mathbf{u}, \theta_c^m p(x) \rangle,$$

understanding that $\theta_c^m p(x) = \theta_c(\theta_c^{m-1} p(x))$.

**Definition 3.** *Let $\mathbf{u}$ be a linear functional and $p(x) = \sum_{k=0}^m a_k x^k$ be a polynomial, and then we define the polynomial $(\mathbf{u} * p)(x)$ as*

$$(\mathbf{u} * p)(x) =: \left\langle \mathbf{u}_y, \frac{xp(x) - yp(y)}{x - y} \right\rangle = \sum_{k=0}^m \left( \sum_{n=k}^m a_n \mathbf{u}_{n-k} \right) x^k.$$

Here, $\mathbf{u}_y$ means that the linear functional $\mathbf{u}$ acts on the variable $y$.

**Definition 4** (Reference [6]). *The product of two linear functionals $\mathbf{u}$ and $\mathbf{v}$ is defined from their moments as follows:*

$$(\mathbf{uv})_n = \langle \mathbf{uv}, x^n \rangle = \sum_{k=0}^n \mathbf{u}_k \mathbf{v}_{n-k}, \quad n \geq 0.$$

*The above product is commutative, associative, and distributive with respect to the sum of linear functionals.*

Let $c$ be a complex number and let $\delta_c$ be the linear functional defined by

$$\langle \delta_c, x^n \rangle = c^n, \quad n \in \mathbb{N}.$$

Notice that, for any linear functional, $\mathbf{u}$, $\mathbf{u}\delta_0 = \mathbf{u}$. Moreover, if the first moment of $\mathbf{u}$ is nonzero, then there exists a unique linear functional $\mathbf{u}^{-1}$ such that $\mathbf{u}\mathbf{u}^{-1} = \delta_0$. The moments of $\mathbf{u}^{-1}$ are defined recursively by

$$(\mathbf{u}^{-1})_n = -\frac{1}{\mathbf{u}_0} \sum_{k=0}^{n-1} \mathbf{u}_{n-k}(\mathbf{u}^{-1})_k, \quad n \geq 1, \quad (\mathbf{u}^{-1})_0 = \mathbf{u}_0^{-1}.$$

**Proposition 1** (**Reference [6]**). *Let $\mathbf{u}, \mathbf{v}$ be linear functionals and $p(x)$ and $q(x)$ polynomials. The following properties hold.*

*(i)*      $\langle \mathbf{uv}, p(x) \rangle = \langle \mathbf{v}, (\mathbf{u} * p)(x) \rangle.$
*(ii)*     $\theta_c(pq)(x) = q(x)\theta_c p(x) + p(c)(\theta_c q(x)).$
*(iii)*    $p^2(x)\mathbf{u}^2 = (p\mathbf{u})^2 + 2xp(x)(\mathbf{u} * \theta_0 p)(x)\mathbf{u}.$

*(iv)*　　　$p(x)(\mathbf{u}\mathbf{v}) = (p(x)\mathbf{v})\mathbf{u} + x(\mathbf{v} * \theta_0 p)(x)\mathbf{u}.$

From Favard's Theorem [2], we know that there exists a unique quasi-definite linear functional $\mathbf{u}$, with $(P_n)_{n\geq 0}$ its corresponding SMOP, if and only if there exist two sequences of complex numbers $(a_n)_{n\geq 1}$ and $(b_n)_{n\geq 0}$, with $a_n \neq 0, n \geq 1$, such that

$$
\begin{aligned}
x\,P_n(x) &= P_{n+1}(x) + b_n\,P_n(x) + a_n\,P_{n-1}(x), \quad n \geq 0, \\
P_{-1}(x) &= 0, \quad P_0(x) = 1.
\end{aligned} \tag{1}
$$

The above recurrence relation can be expressed in matrix form as follows. If $\mathbf{P} = (P_0, P_1, \cdots)^\top$, then $x\mathbf{P} = J\mathbf{P}$, where $A^\top$ denotes the transposed of the matrix $A$, and $J$ is the following tridiagonal semi-infinite matrix (monic Jacobi matrix; see Reference [2])

$$
J = \begin{pmatrix}
b_0 & 1 & & \\
a_1 & b_1 & 1 & \\
& a_2 & b_2 & \ddots \\
& & \ddots & \ddots
\end{pmatrix}.
$$

**Definition 5.** *For $k \in \mathbb{N}$, we define the associated polynomials of the k-th kind $(P_n^{(k)})_{n\geq 0}$, (also called k-th associated polynomials; see Reference [2]) as the sequence of monic polynomials satisfying the recurrence relation*

$$
\begin{aligned}
xP_n^{(k)}(x) &= P_{n+1}^{(k)}(x) + b_{n+k}P_n^{(k)}(x) + a_{n+k}P_{n-1}^{(k)}(x), \quad n \geq 0, \\
P_{-1}^{(k)}(x) &= 0, \quad P_0^{(k)}(x) = 1.
\end{aligned}
$$

*This means that a shift is introduced in the coefficients of the three term recurrence relation (1). According to Favard's Theorem, there exists a quasi-definite linear functional $\mathbf{u}^{(k)}$, called the k-associated transformation of $\mathbf{u}$, such that $(P_n^{(k)})_{n\geq 0}$ is its corresponding SMOP.*

There is a direct representation of such polynomials as (see References [17,18])

$$
P_{n-k}^{(k)}(x) = \frac{1}{\left\langle \mathbf{u}, P_{k-1}^2 \right\rangle} \left\langle P_{k-1}(y)\mathbf{u}_y, \frac{P_n(x) - P_n(y)}{x - y} \right\rangle, \quad n \geq k.
$$

Since $(P_n(x))_{n\geq 0}$ and $(P_{n-1}^{(1)}(x))_{n\geq 0}$ are two linearly independent solutions of the difference equation [18]

$$
xw_n = w_{n+1} + b_n w_n + a_n w_{n-1}, \quad n \geq 1,
$$

every solution can be represented as a linear combination of $(P_n(x))_{n\geq 0}$ and $(P_{n-1}^{(1)}(x))_{n\geq 0}$. In particular (see References [18,19]),

$$
P_{n-k}^{(k)}(x) = A(x,k)P_n(x) + B(x,k)P_{n-1}^{(1)}(x), \quad n \geq k, \tag{2}
$$

where

$$
A(x,k) = -\frac{P_{k-2}^{(1)}(x)}{\prod_{m=1}^{k-1} a_m} \quad \text{and} \quad B(x,k) = \frac{P_{k-1}(x)}{\prod_{m=1}^{k-1} a_m}.
$$

**Definition 6.** *Let $(P_n)_{n\geq 0}$ be a SMOP with respect to $\mathbf{u}$ satisfying the recurrence relation (1). The sequence of monic polynomials $(P_n(x; \mathsf{a}))_{n\geq 0}$ is said to be co-recursive of parameter $\mathsf{a}$ with*

*respect to the linear functional* **u**, *if they also satisfy* (1) *but with initial conditions* $P_0(x; \mathsf{a}) = 1$ *and* $P_1(x; \mathsf{a}) = P_1(x) - \mathsf{a}$. *Notice that*

$$P_n(x; \mathsf{a}) = P_n(x) - \mathsf{a} P_{n-1}^{(1)}(x), \quad n \geq 0.$$

For co-recursive polynomials $P_n(x, \mathsf{a})$, the following three-term recurrence relation holds:

$$\begin{aligned} x\, P_n(x; \mathsf{a}) &= P_{n+1}(x; \mathsf{a}) + b_n\, P_n(x; \mathsf{a}) + a_n\, P_{n-1}(x; \mathsf{a}), \quad n \geq 1, \\ x\, P_0(x; \mathsf{a}) &= P_1(x; \mathsf{a}) + (b_0 + \mathsf{a})\, P_0(x; \mathsf{a}). \end{aligned} \tag{3}$$

**Definition 7.** *Given a quasi-definite linear functional* **u**, *we can define the formal series*

$$S_{\mathbf{u}}(z) =: \sum_{n=0}^{\infty} \frac{\mathbf{u}_n}{z^{n+1}}.$$

*It is said to be the Stieltjes function associated with* **u**.

**Definition 8 (Reference [20]).** *Let* $\widetilde{\mathbf{u}}$ *be a quasi-definite linear functional and* $\widetilde{S}(z)$ *its Stieltjes function.* $\widetilde{\mathbf{u}}$ *is said to be a rational spectral transformed of* **u** *if there exist polynomials* $A(z)$, $B(z)$, $C(z)$, *and* $D(z)$ *such that*

$$\widetilde{S}(z) = \frac{A(z)S_{\mathbf{u}}(z) + B(z)}{C(z)S_{\mathbf{u}}(z) + D(z)}, \quad A(z)D(z) - B(z)C(z) \neq 0.$$

*The above mapping between two linear functionals is called rational spectral transformation. In particular, if* $C(z) \equiv 0$, *then* $\widetilde{\mathbf{u}}$ *is said to be a linear spectral transformed of the linear functional* **u**. *In such a case, the mapping between two linear functionals is called linear spectral transformation.*

**Theorem 1 (References [2,5,6,21,22]).** *Let* $S_{\mathbf{u}}(z)$, $S_{\mathbf{u}^{-1}}(z)$, $S_{\mathbf{u}^{(1)}}(z)$, *and* $S_{\mathbf{u}^{\mathsf{a}}}(z)$ *be the Stieltjes functions associated with* **u**, $\mathbf{u}^{-1}$, $\mathbf{u}^{(1)}$, *and* $\mathbf{u}^{\mathsf{a}}$, *respectively. Then, the following relations hold:*

*(i)*

$$S_{\mathbf{u}}(z)S_{\mathbf{u}^{-1}}(z) = 1/z^2.$$

*(ii)*

$$S_{\mathbf{u}^{(1)}}(z) = -\frac{\mathbf{u}_0 \mathbf{u}_0^{(1)}}{a_1} z^2 S_{\mathbf{u}^{-1}}(z) + \frac{\mathbf{u}_0^{(1)}}{a_1}(z - b_0).$$

*(iii)*

$$S_{\mathbf{u}^{\mathsf{a}}}(z)\left[ \frac{-\mathsf{a}}{(\mathbf{u}^{\mathsf{a}})_0 z^2} + \frac{\mathbf{u}_0}{(\mathbf{u}^{\mathsf{a}})_0} S_{\mathbf{u}^{-1}}(z) \right] = \frac{1}{z^2}.$$

*(iv)*

$$S_{\mathbf{u}}(z) = \frac{\mathbf{u}_0}{(z - b_0) - \dfrac{a_1}{\mathbf{u}_0^{(1)}} S_{\mathbf{u}^{(1)}}(z)}. \tag{4}$$

*Moreover, from the above equations, we can deduce the following relations between the corresponding linear functionals.*

$$\mathbf{u}^{(1)} = -\frac{\mathbf{u}_0^{(1)} \mathbf{u}_0}{a_1} x^2 \mathbf{u}^{-1}, \qquad\qquad \mathbf{u}^{\mathsf{a}} = \frac{\mathbf{u}_0^{\mathsf{a}}}{\mathbf{u}_0}\left( \mathbf{u}^{-1} + \frac{\mathsf{a}}{\mathbf{u}_0} \delta_0' \right)^{-1}. \tag{5}$$

As we have stated before, the analysis of perturbations of linear functionals constitutes an interesting topic in the theory of orthogonal polynomials on the real line (scalar OPRL) (References [14,15,20,23,24] and references therein). Among the perturbations of linear

functionals, spectral linear perturbations have attracted the interest of researchers (see Reference [20]). Such perturbations are generated by two particular families, the so-called Christoffel and Geronimus transformations.

Christoffel perturbations, that appear when considering orthogonality with respect to a new linear functional $\widetilde{\mathbf{u}} = p(x)\mathbf{u}$, where $p(x)$ is a polynomial, were studied in 1858 by E. B. Christoffel (see References [25,26]) when $\mathbf{u}$ is the linear functional associated with the Lebesgue measure $d\mu$ supported on the interval $(-1,1)$ and $d\widehat{\mu}(x) = p(x)d\mu(x)$, with $p(x) = (x - q_1) \cdots (x - q_N)$, a signed polynomial in the support of $d\mu$. Connection formulas between the corresponding SMOP are obtained therein. The location of their zeros as nodes of the quadrature rules is also deduced. More recently, from a numerical point of view, in Reference [27], the authors focus the attention on the sensitivity of Gauss–Christoffel quadrature with respect to small perturbations of the probability measure. On the other hand, the relations between the coefficients of the three term recurrence relations of the corresponding SMOP have been extensively studied (see Reference [28]), as well as the relation between the Jacobi matrices in the framework of the so-called discrete Darboux transformations. They are based on the *LU* factorization of such matrices (see References [14,29], among others).

Notice that the zeros of orthogonal polynomials with respect to *a canonical Christoffel transformation* (a perturbation by a linear polynomial $p(x) = (x - c)$) of a nontrivial probability measure) are the nodes in the Gauss-Radau quadrature formula. In the case of a perturbation of the measure by a positive quadratic polynomial on the support of the measure, the zeros of the corresponding orthogonal polynomials are the nodes of the Gauss-Lobatto quadrature rule (see Reference [13]).

Geronimus transformations appear when you deal with perturbed functionals $\widehat{\mathbf{u}}$ defined by $p(x)\widehat{\mathbf{u}} = \mathbf{u}$, where $p(x)$ is a polynomial, and $\mathbf{u}$ is a quasi-definite linear functional. Such a kind of transformations were used by J. L. Geronimus (see Reference [30]), in order to provide an alternative proof of a result given by W. Hahn [31] concerning the characterization of classical orthogonal polynomials (Hermite, Laguerre, Jacobi, and Bessel) as the unique families of orthogonal polynomials whose first derivatives are also orthogonal polynomials. Examples of such transformations have been done by P. Maroni [32] for a perturbation of the type $p(x) = x - c$, also known as *a canonical Geronimus transformation*. In a similar way, examples for the quadratic and cubic case can be found in References [16,33–35], respectively.

In Reference [22], we study the following problem:

**Problem 1.** *(Figure 1) Let* $\mathbf{u}$ *be a quasi-definite functional and let* $\widetilde{\mathbf{u}} = (x - c)\mathbf{u}$ *and* $(x - c)\widehat{\mathbf{u}} = \mathbf{u}$ *be a canonical Christoffel and Geronimus transformation of* $\mathbf{u}$*, respectively. What is the relation between* $\mathbf{u}^{(1)}$ *and* $\widetilde{\mathbf{u}}^{(1)}$ *(resp.* $\widehat{\mathbf{u}}^{(1)}$*)? There,* $\widetilde{\mathbf{u}}^{(1)}$ *(resp.* $\widehat{\mathbf{u}}^{(1)}$*) is the associated linear functional of the first kind of* $\widetilde{\mathbf{u}}$ *(resp.* $\widehat{\mathbf{u}}$*).*

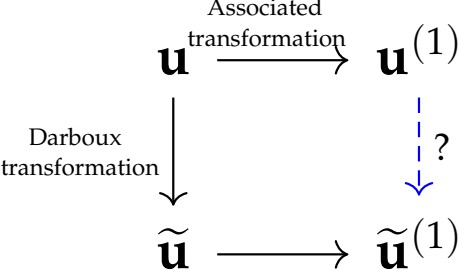

**Figure 1.** Structure of Problem 1.

To give a solution of the above problem, we use the LU and UL factorization of the monic Jacobi matrix associated with $\mathbf{u}$ as well as the co-recursive polynomials.

In the present contribution, we are interested to study the following problem.

**Problem 2.** *(Figure 2) Let* **u** *be a quasi-definite functional, and let* $\widetilde{\mathbf{u}} = (x - c)\mathbf{u}$ *and* $(x - c)\widehat{\mathbf{u}} = \mathbf{u}$ *be a canonical Christoffel and Geronimus transformation of* **u**, *respectively. What is the relation between* $\widetilde{\mathbf{u}}$ *and* $\widetilde{\mathbf{u}^{(1)}}$ *(resp.* $\widehat{\mathbf{u}^{(1)}}$*)? There,* $\widetilde{\mathbf{u}^{(1)}}$ *(resp.* $\widehat{\mathbf{u}^{(1)}}$*) is the Christoffel (resp. Geronimus) transformation of* $\mathbf{u}^{(1)}$.

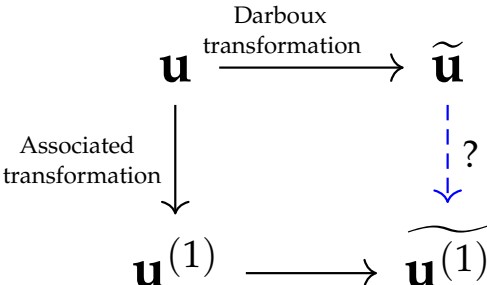

**Figure 2.** Structure of Problem 2.

　　With this in mind, the structure of the manuscript is as follows. In Section 2, for a canonical Christoffel transformation, we study the connection between the linear functionals $\widetilde{\mathbf{u}}$ and $\widetilde{\mathbf{u}^{(1)}}$. As a consequence, we deduce the relation between the corresponding Stieltjes functions, as well as the sequences of monic orthogonal polynomials. In a second step, we study the same problem when a canonical Geronimus transformation is considered, i.e., the relation between $\widehat{\mathbf{u}}$ and $\widehat{\mathbf{u}^{(1)}}$. In Section 3, we analyze a general family of linear functionals (the so-called Laguerre-Hahn) whose Stieltjes function satisfies a Riccati equation. Once we introduce the definition of the class of such a linear functional, we study the class of a Laguerre-Hahn linear functional when either a canonical Christoffel or a Geronimus transformation is implemented. Finally, two illustrative examples about the above questions are discussed.

## 2. Darboux Transformation and Associated Polynomials of the First Kind

### 2.1. Christoffel Transformation and Its Associated Polynomials of First Kind

　　Let **u** be a quasi-definite linear functional, and let $(P_n(x))_{n \geq 0}$ be its corresponding SMOP. If $c$ is a complex number, the linear functional $\widetilde{\mathbf{u}} = (x - c)\mathbf{u}$ is said to be a canonical Christoffel transformation of the functional **u**. Let us assume that the linear functional $\widetilde{\mathbf{u}}$ is also quasi-definite which is equivalent to $P_n(c) \neq 0$ for all $n \in \mathbb{N}$, and let $(\widetilde{P}_n)_{n \geq 0}$ be its SMOP. It is well known that $(P_n)_{n \geq 0}$ and $(\widetilde{P}_n)_{n \geq 0}$ are related by

$$(x - c)\widetilde{P}_n(x) = P_{n+1}(x) - \frac{P_{n+1}(c)}{P_n(c)}P_n(x), \quad n \geq 0.$$

We have the following relation between their Jacobi matrices.

**Theorem 2 (References [14,29]).** *Let $J$ and $\widetilde{J}$ be the Jacobi matrices associated with* **u** *and* $\widetilde{\mathbf{u}} = (x - c)\mathbf{u}$, *respectively. If $P_n(c) \neq 0$, for all $n \in \mathbb{N}$, then $J - cI$ has a LU factorization, i.e.,*

$$J - cI := LU := \begin{pmatrix} 1 & & & \\ \ell_1 & 1 & & \\ & \ell_2 & 1 & \\ & & \ddots & \ddots \end{pmatrix}\begin{pmatrix} \beta_0 & 1 & & & \\ & \beta_1 & 1 & & \\ & & \beta_2 & \ddots & \\ & & & \ddots & \ddots \end{pmatrix},$$

*where L is a lower bidiagonal matrix with 1's in the main diagonal, and U is an upper bidiagonal matrix with $\beta_n = -P_{n+1}(c)/P_n(c)$.*

$$\begin{cases} b_n - c = \ell_n + \beta_n, & b_0 - c = \beta_0, \\ a_n = \ell_n \beta_{n-1}, & n = 1, 2 \ldots \end{cases}$$

*Moreover, $\widetilde{J} - cI = UL$, where*

$$
\begin{cases}
\widetilde{b}_n - c = \ell_{n+1} + \beta_n, \\
\widetilde{a}_n = \ell_n \beta_n,
\end{cases}
\quad n = 0, 1, 2 \dots
$$

**Proposition 2 (Reference [20]).** *Let $S_{\mathbf{u}}(z)$ and $S_{\widetilde{\mathbf{u}}}(z)$ be the Stieltjes functions associated with $\mathbf{u}$ and $\widetilde{\mathbf{u}}$, respectively. Then, $S_{\widetilde{\mathbf{u}}}(z)$ is a linear spectral transformation of $S_{\mathbf{u}}(z)$. Indeed, the moments of $\widetilde{\mathbf{u}}$ and $\mathbf{u}$ satisfy the following relation*

$$
\widetilde{\mathbf{u}}_n = \langle \widetilde{\mathbf{u}}, x^n \rangle = \langle \mathbf{u}, (x - c)\, x^n \rangle = \mathbf{u}_{n+1} - c\, \mathbf{u}_n.
$$

*From here,*

$$
S_{\widetilde{\mathbf{u}}}(z) = (z - c)\, S_{\mathbf{u}}(z) - \mathbf{u}_0. \tag{6}
$$

Since $\mathbf{u}$ is a quasi-definite linear functional, then $\mathbf{u}^{(1)}$ is quasi-definite, Let $(P_n(x))_{n \geq 0}$ be the SMOP with respect to $\mathbf{u}$, and assume that $P_n(c) \neq 0$ for every $n \in \mathbb{N}$. We are interested to analyze the relation between the linear functionals $\widetilde{\mathbf{u}}$ and $\widetilde{\mathbf{u}^{(1)}}$ given by $\widetilde{\mathbf{u}^{(1)}} := (x - c)\mathbf{u}^{(1)}$ and $\widetilde{\mathbf{u}} := (x - c)\mathbf{u}$, respectively.

**Proposition 3.** *The linear functionals $\widetilde{\mathbf{u}}$ and $\widetilde{\mathbf{u}^{(1)}}$ are related as follows:*

$$
\widetilde{\mathbf{u}^{(1)}} = s(x - c)(x - t)\left( (\widetilde{\mathbf{u}})^{-1} - \frac{1}{\mathbf{u}_0} \delta_0' \right)^{-1}, \tag{7}
$$

*with $t = \widetilde{a}_1 \dfrac{P_1(c)}{P_2(c)} + \widetilde{b}_0 - \dfrac{\widetilde{\mathbf{u}}_0}{\mathbf{u}_0}$ and $s = -\dfrac{P_2(c)}{P_1(c)} \dfrac{\mathbf{u}_0^{(1)}}{\widetilde{a}_1 \widetilde{\mathbf{u}}_0}$.*

**Proof.** In (Reference [22], Proposition 16), the following relation was proven.

$$
\widetilde{P}_n(x) + \frac{\widetilde{\mathbf{u}}_0}{\mathbf{u}_0} \widetilde{P}_{n-1}^{(1)}(x) = P_n^{(1)}(x) - \frac{P_{n+1}(c)}{P_n(c)} P_{n-1}^{(1)}(x), \quad n \geq 0, \tag{8}
$$

where $(\widetilde{P}_n^{(1)})_{n \geq 0}$ is the SMOP of the first kind associated transformation of the linear functional $\widetilde{\mathbf{u}}$ (see Definition 5). Notice that the polynomials in the left-hand side of (8) are co-recursive of parameter $a = -\frac{\widetilde{\mathbf{u}}_0}{\mathbf{u}_0}$ with respect to the linear functional $\widetilde{\mathbf{u}}$. Let us denote

$$
V_n(x) = \widetilde{P}_n(x) + \frac{\widetilde{\mathbf{u}}_0}{\mathbf{u}_0} \widetilde{P}_{n-1}^{(1)}(x), \quad n \geq 0,
$$

such a monic polynomial sequence, and let $\mathbf{w}$ be the linear functional such that $(V_n)_{n \geq 0}$ is the corresponding SMOP. Then, from (5),

$$
\mathbf{w} = \frac{\mathbf{w}_0}{\widetilde{\mathbf{u}}_0}\left( (\widetilde{\mathbf{u}})^{-1} - \frac{1}{\mathbf{u}_0} \delta_0' \right)^{-1}. \tag{9}
$$

If we expand the linear functional $\mathbf{u}^{(1)}$ in the dual basis

$$
\left( \frac{V_n(x)\mathbf{w}}{\langle \mathbf{w}, V_n^2(x) \rangle} \right)_{n \geq 0}
$$

of the polynomials $(V_n(x))_{n \geq 0}$ [6] and using the fact that $\left\langle \mathbf{u}^{(1)}, V_n(x) \right\rangle = 0$ for all $n \geq 2$, we get

$$\mathbf{u}^{(1)} = \alpha_0 \frac{\mathbf{w}}{\mathbf{w}_0} + \alpha_1 \frac{V_1(x)\mathbf{w}}{\langle \mathbf{w}, V_1^2(x) \rangle}.$$

From the orthogonal relations, we obtain $\alpha_0 = \mathbf{u}_0^{(1)}$, $\alpha_1 = -\frac{P_2(c)}{P_1(c)} \mathbf{u}_0^{(1)}$. Thus,

$$\mathbf{u}^{(1)} = -\frac{P_2(c)}{P_1(c)} \frac{\mathbf{u}_0^{(1)}}{\langle \mathbf{w}, V_1(x) \rangle} \left( V_1(x) - \frac{P_1(c)}{P_2(c)} \frac{\langle \mathbf{w}, V_1(x) \rangle}{\mathbf{w}_0} \right) \mathbf{w}.$$

Taking into account that $\langle \mathbf{w}, V_1(x) \rangle = \widetilde{a}_1 \mathbf{w}_0$ and (9), we obtain the result. $\square$

Let us assume that $P_n^{(1)}(c) \neq 0$ for all $n \in \mathbb{N}$. Then, the SMOP $(\widetilde{P_n^{(1)}}(x))_{n \geq 0}$ with respect to $\widetilde{\mathbf{u}^{(1)}}$ satisfies

$$(x - c)\widetilde{P_n^{(1)}}(x) = P_{n+1}^{(1)}(x) - \frac{P_{n+1}^{(1)}(c)}{P_n^{(1)}(c)} P_n^{(1)}(x), \quad n \geq 0.$$

An equivalent condition is given in terms of the co-recursive polynomials $(V_n(x))_{n \geq 0}$ defined in the above Proposition. Indeed:

**Corollary 1.** $\widetilde{\mathbf{u}^{(1)}}$ *is quasi-definite if and only if $d_n \neq 0$ for every $n \in \mathbb{N}$, where*

$$d_n = \begin{cases} \det \begin{pmatrix} V_{n+1}(c) & V_n(c) \\ V_{n+1}(t) & V_n(t) \end{pmatrix}, & \text{if } t \neq c, \\[12pt] \det \begin{pmatrix} V_{n+1}(c) & V_n(c) \\ V'_{n+1}(c) & V'_n(c) \end{pmatrix}, & \text{if } t = c, \end{cases} \quad \text{with} \quad V_n(x) = \widetilde{P}_n(x) + \frac{\widetilde{\mathbf{u}}_0}{\mathbf{u}_0} \widetilde{P}_{n-1}^{(1)}(x).$$

*Moreover,*

$$(x - c)(x - t)\widetilde{P_n^{(1)}}(x) = \begin{cases} \dfrac{1}{d_n} \det \begin{pmatrix} V_{n+2}(x) & V_{n+1}(x) & V_n(x) \\ V_{n+2}(c) & V_{n+1}(c) & V_n(c) \\ V_{n+2}(t) & V_{n+1}(t) & V_n(t) \end{pmatrix}, & \text{if } t \neq c, \\[24pt] \dfrac{1}{d_n} \det \begin{pmatrix} V_{n+2}(x) & V_{n+1}(x) & V_n(x) \\ V_{n+2}(c) & V_{n+1}(c) & V_n(c) \\ V'_{n+2}(c) & V'_{n+1}(c) & V'_n(c) \end{pmatrix}, & \text{if } t = c, \end{cases} \quad n \geq 0,$$

*where $t$ is as in Proposition 3.*

**Proof.** The proof is a consequence of (7). $\square$

**Proposition 4.** *The Stieltjes functions $S_{\widetilde{\mathbf{u}}}(z)$ and $S_{\widetilde{\mathbf{u}^{(1)}}}(z)$ are related as follows.*

$$S_{\widetilde{\mathbf{u}^{(1)}}}(z) = \frac{A(z)\, S_{\widetilde{\mathbf{u}}}(z) + B(z)}{S_{\widetilde{\mathbf{u}}}(z) + \mathbf{u}_0},$$

*where*

$$A(z) = \frac{\mathbf{u}_0^{(1)}}{a_1}[(z - c)(z - b_0) - a_1], \quad B(z) = \frac{\mathbf{u}_0 \mathbf{u}_0^{(1)}}{a_1}[(c - b_0)(z - c) - a_1].$$

**Proof.** From (4) and (6),

$$
S_{\widetilde{\mathbf{u}^{(1)}}}(z) = (z-c)S_{\mathbf{u}^{(1)}}(z) - \mathbf{u}_0^{(1)}
$$

$$
= (z-c)\frac{\mathbf{u}_0^{(1)}}{a_1}\left[(z-b_0) - \mathbf{u}_0\frac{(z-c)}{S_{\widetilde{\mathbf{u}}}(z) + \mathbf{u}_0}\right] - \mathbf{u}_0^{(1)}
$$

$$
= \frac{(z-c)\dfrac{\mathbf{u}_0^{(1)}}{a_1}[(z-b_0)S_{\widetilde{\mathbf{u}}}(z) + (z-b_0)\mathbf{u}_0 - \mathbf{u}_0(z-c)] - \mathbf{u}_0^{(1)}(S_{\widetilde{\mathbf{u}}}(z) + \mathbf{u}_0)}{S_{\widetilde{\mathbf{u}}}(z) + \mathbf{u}_0}
$$

$$
= \frac{A(z)\,S_{\widetilde{\mathbf{u}}}(z) + B(z)}{S_{\widetilde{\mathbf{u}}}(z) + \mathbf{u}_0}.
$$

□

*2.2. Geronimus Transformation and Its Associated Polynomials of the First Kind*

Let **v** be a quasi-definite linear functional, and let $(P_n)_{n\geq 0}$ be its corresponding SMOP.

**Definition 9.** *Given a complex number c, a linear functional $\widehat{\mathbf{v}}$ defined by $(x-c)\widehat{\mathbf{v}} = \mathbf{v}$ is said to be a canonical Geronimus transformation of the linear functional* **v**.

It is important to emphasize that $\widehat{\mathbf{v}}$ is not uniquely defined since its first moment is arbitrary. The explicit expression of $\widehat{\mathbf{v}}$ is [29]

$$
\widehat{\mathbf{v}} = (x-c)^{-1}\mathbf{v} + \widehat{\mathbf{v}}_0\delta_c.
$$

If we assume that $\widehat{\mathbf{v}}$ is also quasi-definite, and $(\widehat{P}_n)_{n\geq 0}$ is its corresponding SMOP, then it is well known that $(P_n)_{n\geq 0}$ and $(\widehat{P}_n)_{n\geq 0}$ are related by [14,29]

$$
\widehat{P}_n(x) = P_n(x) + \ell_n P_{n-1}(x), \quad n \geq 1,
$$

where

$$
\ell_n = -\frac{\mathbf{v}_0 P_{n-1}^{(1)}(c) + \widehat{\mathbf{v}}_0 P_n(c)}{\mathbf{v}_0 P_{n-2}^{(1)}(c) + \widehat{\mathbf{v}}_0 P_{n-1}(c)}, \quad n \geq 1. \tag{10}
$$

Thus, a necessary and sufficient condition on $\widehat{\mathbf{v}}$ to be a quasi-definite linear functional is

$$
\widehat{\mathbf{v}}_0 P_n(c) + \mathbf{v}_0 P_{n-1}^{(1)}(c) \neq 0, \quad \text{for all } n \geq 1. \tag{11}
$$

A second relation between $(P_n)_{n\geq 0}$ and $(\widehat{P}_n)_{n\geq 0}$ is (see Reference [29])

$$
(x-c)P_n(x) = \widehat{P}_{n+1}(x) + \beta_n\widehat{P}_n(x), \quad n \geq 0, \tag{12}
$$

where $\beta_n = -\widehat{P}_{n+1}(c)/\widehat{P}_n(c)$.

**Theorem 3 (Reference [20]).** *If $S_{\mathbf{v}}(z)$ and $S_{\widehat{\mathbf{v}}}(z)$ are the Stieltjes functions for* **v** *and* $\widehat{\mathbf{v}}$, *respectively, then $S_{\widehat{\mathbf{v}}}(z)$ is a linear spectral transformation of $S_{\mathbf{v}}(z)$. Indeed, taking into account that the moments of $\widehat{\mathbf{v}}$ and* **v** *satisfy*

$$
\mathbf{v}_n = \langle \mathbf{v}, x^n \rangle = \langle \widehat{\mathbf{v}}, (x-a)\,x^n \rangle = \widehat{\mathbf{v}}_{n+1} - c\,\widehat{\mathbf{v}}_n,
$$

*then*

$$
S_{\widehat{\mathbf{v}}}(z) = \frac{S_{\mathbf{v}}(z) + \widehat{\mathbf{v}}_0}{(z-c)}.
$$

Returning to the previous discussion, (10) and (12) imply the following relation between the corresponding monic Jacobi matrices associated with **v** and $\widehat{\mathbf{v}}$.

**Theorem 4** (**References [14,23,24,29]**). *Let $J$ and $\widehat{J}$ be the monic Jacobi matrices associated with* **v** *and* $\widehat{\mathbf{v}}$, *respectively. If* $\widehat{\mathbf{v}}_0$ *satisfies* (11), *then* $J - cI$ *has an UL factorization. Indeed,*

$$J - cI := UL := \begin{pmatrix} \beta_0 & 1 & & & \\ & \beta_1 & 1 & & \\ & & \beta_2 & \ddots & \\ & & & \ddots & \ddots \end{pmatrix} \begin{pmatrix} 1 & & & \\ \ell_1 & 1 & & \\ & \ell_2 & 1 & \\ & & \ddots & \ddots \end{pmatrix}, \tag{13}$$

*or, equivalently,*

$$\begin{cases} b_n - c = \ell_{n+1} + \beta_n, & n = 0, 1, \dots \\ a_n = \ell_n \beta_n, & n = 1, 2 \dots \end{cases}$$

*where $L$ is a lower bidiagonal matrix with 1's as diagonal entries, and $U$ is an upper bidiagonal matrix with $\beta_n = -\widehat{P}_{n+1}(c)/\widehat{P}_n(c)$. Moreover,*

$$\widehat{J} - cI = LU.$$

*If we assume that $\ell_0 := 0$, then the corresponding entries satisfy*

$$\begin{cases} \widehat{b}_n - c = \ell_n + \beta_n, & n = 0, 1, \dots \\ \widehat{a}_n = \ell_n \beta_{n-1}, & n = 1, 2 \dots \end{cases}$$

Observe that the UL factorization depends on the choice of $\widehat{\mathbf{v}}_0$ since $\beta_0 = \mathbf{v}_0 / \widehat{\mathbf{v}}_0$.

The matrices $U$ and $L$ given in (13) can be written

$$U = \left( \begin{array}{c|ccc} \beta_0 & 1 & 0 & \cdots \\ \hline 0 & & & \\ 0 & & U_1 & \\ \vdots & & & \end{array} \right), \quad L = \left( \begin{array}{c|ccc} 1 & 0 & 0 & \cdots \\ \hline \ell_1 & & & \\ 0 & & L_1 & \\ \vdots & & & \end{array} \right),$$

where $U_1$ and $L_1$ are upper and lower bidiagonal matrices, respectively. From here, we deduce that the semi infinite matrix $J^{(1)} - cI$, with $J^{(1)}$ the Jacobi matrix associated with $\mathbf{v}^{(1)}$, has also an UL factorization, i.e.,

$$J^{(1)} - cI := U_1 L_1 := \begin{pmatrix} \beta_1 & 1 & & & \\ & \beta_2 & 1 & & \\ & & \beta_3 & \ddots & \\ & & & \ddots & \ddots \end{pmatrix} \begin{pmatrix} 1 & & & \\ \ell_2 & 1 & & \\ & \ell_3 & 1 & \\ & & \ddots & \ddots \end{pmatrix}.$$

Moreover, since $\beta_1 = \dfrac{a_1}{b_0 - c - \beta_0}$, then $\beta_1$ also depends on the choice of $\widehat{\mathbf{v}}_0$. Now, if we define

$$\widehat{J^{(1)}} - cI := L_1 U_1 = \begin{pmatrix} \beta_1 & 1 & & \\ \widehat{a}_2 & \widehat{b}_2 - c & 1 & \\ & \widehat{a}_3 & \widehat{b}_3 - c & \ddots \\ & & \ddots & \ddots \end{pmatrix}, \tag{14}$$

then $\widehat{J^{(1)}}$ is the Jacobi matrix associated with the linear functional $\widehat{\mathbf{u}^{(1)}}$ defined by

$$\widehat{\mathbf{u}^{(1)}} = (x - c)^{-1} \mathbf{u}^{(1)} + \widehat{\mathbf{u}_0^{(1)}} \delta_c, \quad \text{where} \quad \widehat{\mathbf{u}_0^{(1)}} = \mathbf{u}_0^{(1)} / \beta_1. \tag{15}$$

Let $(\widehat{P_n^{(1)}})_{n\geq 0}$ be the SMOP with respect to $\widehat{\mathbf{u}^{(1)}}$. Then, from (14), we can deduce the following.

**Proposition 5.** *Let $(\widehat{P}_n^{(1)})_{n\geq 0}$ be the SMOP with respect to the first kind associated transformation of the linear functional $\widehat{\mathbf{v}}$ that will be denoted by $\mathbf{w}$. Then, the polynomials $(\widehat{P}_n^{(1)})_{n\geq 0}$ are co-recursive of parameter $\mathsf{a} = \ell_1$ with respect to $\mathbf{w}$, i.e.,*

$$\widehat{P_n^{(1)}}(x) = \widehat{P}_n^{(1)}(x) - \ell_1\widehat{P_{n-1}^{(2)}}(x), \quad n \geq 0. \tag{16}$$

*Moreover, the linear functional $\widehat{\mathbf{v}^{(1)}}$ such that $(\widehat{P_n^{(1)}})_{n\geq 0}$ is the corresponding SMOP can be written as*

$$\widehat{\mathbf{v}^{(1)}} = \frac{\widehat{\mathbf{v}_0^{(1)}}}{\mathbf{w}_0}\left(\mathbf{w}^{-1} - \frac{\ell_1}{\mathbf{w}_0}\delta_0'\right)^{-1}. \tag{17}$$

**Proof.** In Reference [22], Proposition 24, it is proved that, if the Jacobi matrix $J - cI$ has a LU factorization as in (13), then the Jacobi matrix $J_{\mathbf{w}}$ associated with $\mathbf{w}$ satisfies

$$J_{\mathbf{w}} - cI = \begin{pmatrix} \ell_1 + \beta_1 & 1 & & \\ \ell_2\beta_1 & \ell_2 + \beta_2 & 1 & \\ & \ell_3\beta_2 & \ell_3 + \beta_3 & 1 \\ & & \ddots & \ddots & \ddots \end{pmatrix} = \begin{pmatrix} \widehat{b}_1 - c & 1 & & \\ \widehat{a}_2 & \widehat{b}_2 - c & 1 & \\ & \widehat{a}_3 & \widehat{b}_3 - c & \ddots \\ & & \ddots & \ddots \end{pmatrix}. \tag{18}$$

A comparison between the entries of the matrices (18) and (14) and taking into account (3), yield (16). (17) is a direct consequence of (5). $\quad\square$

**Corollary 2.** *Under the hypothesis of Proposition 5, we get the following relations:*

(i). $\widehat{P_n^{(1)}}(x) = \left[1 - \dfrac{\ell_1}{\widehat{a}_1}(x - \widehat{b}_0)\right](\widehat{P}_n)^{(1)}(x) + \dfrac{\ell_1}{\widehat{a}_1}\widehat{P}_{n+1}(x), n \geq 2.$

(ii). $(x - c)^{-1}\mathbf{v}^{(1)} + \widehat{\mathbf{v}_0^{(1)}}\delta_c = \dfrac{\widehat{\mathbf{v}_0^{(1)}}}{\mathbf{w}_0}\left(\mathbf{w}^{-1} - \dfrac{\ell_1}{\mathbf{w}_0}\delta_0'\right)^{-1}.$

**Proposition 6.** *The Stieltjes functions $S_{\widehat{\mathbf{v}}}$ and $S_{\widehat{\mathbf{v}^{(1)}}}$ are related as follows:*

$$S_{\widehat{\mathbf{v}^{(1)}}}(z) = \frac{A(z)S_{\widehat{\mathbf{v}}}(z) + B(z)}{(z - c)^2 S_{\widehat{\mathbf{v}}}(z) + (z - c)\widehat{\mathbf{v}}_0},$$

*where*

$$A(z) = (z - c)\left[\frac{\mathbf{v}_0^{(1)}}{a_1}(z - b_0) + \widehat{\mathbf{v}_0^{(1)}}\right], \qquad B(z) = \frac{\mathbf{v}_0^{(1)}}{a_1}[\widehat{\mathbf{v}}_0(z - b_0) - \mathbf{v}_0] - \widehat{\mathbf{v}}_0\widehat{\mathbf{v}_0^{(1)}}.$$

**Proof.** It follows the guidelines of the proof of Proposition 4. $\quad\square$

## 3. Laguerre-Hahn Linear Functional

**Definition 10 (References [8,9,11]).** *A linear functional $\mathbf{u}$ is said to be of the Laguerre-Hahn class if its Stieltjes function satisfies a Riccati equation*

$$\phi(z)S_{\mathbf{u}}'(z) = A(z)S_{\mathbf{u}}^2(z) + B(z)S_{\mathbf{u}}(z) + C(z), \tag{19}$$

*where $\phi(z) \neq 0$, $A(z)$, $B(z)$, $C(z)$ are polynomials with*

$$C(z) = (D\mathbf{u} * \theta_0\phi)(z) - (\mathbf{u} * \theta_0 B)(z) - (\mathbf{u}^2 * \theta_0^2 A)(z). \tag{20}$$

Recall that $\theta_0$ was given in Definition 2. In particular, if $A(z) \equiv 0$, then the linear functional is said to be semi-classical.

**Proposition 7** (**References [8,9,11]**). *Let* **u** *be a quasi-definite and normalized linear functional, i.e.,* $\mathbf{u}_0 = 1$, *and let* $(P_n)_{n \geq 0}$ *be its corresponding SMOP. The following statements are equivalent*

*(i)* **u** *is a Laguerre-Hahn functional.*

*(ii)* **u** *satisfies the functional equation*

$$\mathcal{D}(\phi(x)\mathbf{u}) + \psi(x)\mathbf{u} - A(x)(x^{-1}\mathbf{u}^2) = 0, \tag{21}$$

*where* $\phi(x)$, $A(x)$, $B(x)$, $C(x)$ *are the polynomials in* (19) *and*

$$\psi(x) = -[\phi'(x) + B(x)].$$

*(iii)* *Each polynomial* $P_n(x)$ *verifies the so-called structure relation*

$$\phi(x)P'_{n+1}(x) + A(x)P_n^{(1)}(x) = \sum_{k=n-s}^{n+d} \lambda_{n,k} P_k(x), \quad n \geq s+1.$$

*Here,* $\phi(x)$ *and* $A(x)$ *are the polynomials given in* (19), $s = \max\{t-1, d-2\}$ *and* $d = \max\{r, m\}$, *where* $r = \deg \phi$, $t = \deg \psi$, *and* $m = \deg A$.

**Remark 1.** *We notice that there are changes of signs in the previous characterizations compared to the works of Maroni [5,6], Belmehdi [7], Marcellán and Prianes [11,12], and many other authors. This is because, in these articles, the Stieltjes function was multiplied by a negative sign.*

In characterization (ii), we must notice that you have not uniqueness in the representation. Indeed, if **u** is Laguerre-Hahn, and $q(x)$ is a polynomial, then **u** also satisfies the functional equation

$$\mathcal{D}(q(x)\phi(x)\mathbf{u}) + (q(x)\psi(x) - q'(x)\phi(x))\mathbf{u} - q(x)A(x)(x^{-1}\mathbf{u}^2) = 0.$$

Notice that the above implies that, if **u** is a Laguerre-Hahn functional, then the Ricatti Equation (19) is not unique. With this in mind, we give the following definition:

**Definition 11** (**References [8,9,11]**). *The class of a Laguerre-Hahn functional* **u** *is the nonnegative integer number defined as*

$$\mathfrak{s} := \min \max\{\deg \psi(x) - 1, \ \max\{\deg \phi(x), \deg A(x)\} - 2\},$$

*where the minimum is taken among all polynomials* $\phi(x)$, $\psi(x)$, *and* $A(x)$ *such that* **u** *satisfies* (21).

Taking into account that the class of a Laguerre-Hahn linear functional is very useful in order to state a hierarchy of such families, we need to give a simple way to characterize it.

**Proposition 8** (**References [9,11]**). *Let* **u** *be a Laguerre-Hahn linear functional, and let* $\phi(x)$ *and* $\psi(x)$ *be non-zero polynomials with* $\deg \phi(x) =: r$, $\deg \psi(x) =: t$ *and* $\deg A(x) =: m$, *such that* (21) *holds. Let* $s := \max\{t-1, d-2\}$ *with* $d = \max\{r, m\}$. *Then,* $s$ *is the class of* **u** ($s = \mathfrak{s}$) *if and only if*

$$\prod_{a:\, \phi(a)=0} \left( |\psi(a) + \phi'(a)| + |A(a)| + \left| \langle \mathbf{u}, \theta_a \psi(x) + \theta_a^2 \phi(x) - (\mathbf{u} * \theta_0[\theta_a A(x)]) \rangle \right| \right) > 0.$$

From the above Theorem, there is an alternative way to find the class in terms of the polynomials involved in the Riccati Equation (19). Indeed:

**Corollary 3 (References [9,11]).** *Let* **u** *be a Laguerre-Hahn functional satisfying* (19) *such that* $\deg \phi(x) = r$, $\deg A(x) = m$ *and* $\deg \psi = t$ *with* $\psi(x) = -[\phi'(x) + B(x)]$. *Let* $s = \max\{t-1, d-2\}$ *with* $d = \max\{r, m\}$. *Then,* $\mathfrak{s} = s$ *if and only if the polynomials* $\phi(x)$, $A(x)$, $B(x)$ *and* $C(x)$ *are coprime or, equivalently,*

$$\prod_{a:\phi(a)=0} \Big( |A(a)| + |B(a)| + |C(a)| \Big) > 0.$$

**Theorem 5.** *Let* $\mathbf{u}^{(1)}$ *be the first associated transformation of* **u** *and assume without lost of generality that* $\mathbf{u}_0 = \mathbf{u}_0^{(1)} = 1$. *If* **u** *is a Laguerre-Hahn functional of class* $\mathfrak{s}$ *satisfying* (19), *then so is* $\mathbf{u}^{(1)}$. *In this case, we have that*

$$\phi_1(z)S'_{\mathbf{u}^{(1)}}(z) = A_1(z)S^2_{\mathbf{u}^{(1)}}(z) + B_1(z)S_{\mathbf{u}^{(1)}}(z) + C_1(z), \tag{22}$$

*where*

$$\begin{aligned}
\phi_1(z) &= \phi(z), \\
A_1(z) &= a_1 \, C(z), \\
B_1(z) &= -2(z - b_0)C(z) - B(z), \\
C_1(z) &= \frac{1}{a_1}\Big[\phi(z) + A(z) + (z - b_0)B(z) + (z - b_0)^2 C(z)\Big].
\end{aligned}$$

*The above polynomials are coprime. Moreover, if* $\mathfrak{s}_1$ *is the class of* $\mathbf{u}^{(1)}$, *then* $\mathfrak{s} - 2 \leq \mathfrak{s}_1 \leq \mathfrak{s}$.

**Proof.** Let $a$ be a zero of $\phi_1(x)$. If $A_1(a) \neq 0$, we get the result. If $A_1(a) = 0$, then $B_1(a) = -B(a)$. If $B(a) \neq 0$, we stop the analysis. If $B(a) = 0$, then $C_1(a) = A(a) \neq 0$ necessarily, since, in other cases, we would have a contradiction with the class of **u**. Thus, we have that $\phi_1$, $A_1$, $B_1$, and $C_1$ are coprime.

Now, since **u** is of class $\mathfrak{s}$, then $\deg \phi \leq \mathfrak{s} + 2$, $\deg A \leq \mathfrak{s} + 2$ and $\deg \psi \leq \mathfrak{s} + 1$. Denote

$$\phi(x) = \sum_{k=0}^{\mathfrak{s}+2} \lambda_k x^k, \qquad B(x) = \sum_{k=0}^{\mathfrak{s}+1} \beta_k x^k, \qquad A(x) = \sum_{k=0}^{\mathfrak{s}+2} \alpha_k x^k. \tag{23}$$

Using (20), we have $\deg C \leq \mathfrak{s}$. Moreover,

$$\begin{aligned}
C(x) := \sum_{k=0}^{\mathfrak{s}} c_k x^k = &-(\lambda_{\mathfrak{s}+2} + \beta_{\mathfrak{s}+1} + \alpha_{\mathfrak{s}+2})x^{\mathfrak{s}} \\
&- (\lambda_{\mathfrak{s}+1} + 2b_0\lambda_{\mathfrak{s}+2} + \beta_{\mathfrak{s}} + \beta_{\mathfrak{s}+1}b_0 + \alpha_{\mathfrak{s}+1} + 2b_0\alpha_{\mathfrak{s}+2})x^{\mathfrak{s}-1} + \cdots.
\end{aligned}$$

On the other hand, taking into account that

$$\begin{aligned}
\psi_1(x) &= -[\phi'(x) - 2(x - b_0)C(x) - B(x)] \\
&= -[(\mathfrak{s}+2)\lambda_{\mathfrak{s}+2} - 2c_{\mathfrak{s}} - \beta_{\mathfrak{s}+1}]x^{\mathfrak{s}+1} - [(\mathfrak{s}+1)\lambda_{\mathfrak{s}+1} - 2c_{\mathfrak{s}-1} + 2b_0c_{\mathfrak{s}} - \beta_{\mathfrak{s}}]x^{\mathfrak{s}} + \cdots,
\end{aligned}$$

then, we can distinguish the following cases:

**(1)** If $\lambda_{\mathfrak{s}+2} \neq 0$, then $\mathfrak{s}_1 = \mathfrak{s}$.

**(2)** If $\lambda_{\mathfrak{s}+2} = 0$ and $2c_{\mathfrak{s}} + \beta_{\mathfrak{s}+1} \neq 0$, then $\mathfrak{s}_1 = \mathfrak{s}$.

**(3)** If $\lambda_{\mathfrak{s}+2} = 0$ and $2c_{\mathfrak{s}} + \beta_{\mathfrak{s}+1} = 0$, we have the subcases:

    **(3-1)** If $\lambda_{\mathfrak{s}+1} \neq 0$, then $\mathfrak{s}_1 = \mathfrak{s} - 1$.

    **(3-2)** If $\lambda_{\mathfrak{s}+1} = 0$ and $2c_{\mathfrak{s}-1} - 2b_0c_{\mathfrak{s}} + \beta_{\mathfrak{s}} \neq 0$, then $\mathfrak{s}_1 = \mathfrak{s} - 1$.

    **(3-3)** If $\lambda_{\mathfrak{s}+1} = 0$ and $2c_{\mathfrak{s}-1} - 2b_0c_{\mathfrak{s}} + \beta_{\mathfrak{s}} = 0$.

        **(3-3-1)** In this case, the leading coefficient of $A_1(x)$ reduces to $a_1c_{\mathfrak{s}} = a_1(\beta_{\mathfrak{s}+1} + \alpha_{\mathfrak{s}+2})$. If zero, then, from item (3), $\beta_{\mathfrak{s}+1} = 0$. In conclusion, we would

have $((\mathfrak{s}+2)\lambda_{\mathfrak{s}+1} + \beta_{\mathfrak{s}+1}) = \lambda_{\mathfrak{s}+2} = \alpha_{\mathfrak{s}+2} = 0$, which is contradictory with the class of **u**. Thus, $a_1 c_{\mathfrak{s}} \neq 0$ and $\mathfrak{s}_1 = \mathfrak{s} - 2$.

$\square$

**Corollary 4.** *With the notation of* (23)*, if* $\mathfrak{s}$ *and* $\mathfrak{s}_1$ *are the classes of* **u** *and* $\mathbf{u}^{(1)}$*, respectively, we get:*

- *If* $|\beta_{s+1} + \alpha_{s+2}| + |\lambda_{s+2}| \neq 0$*, then* $\mathfrak{s}_1 = \mathfrak{s}$*.*
- *If* $|\beta_{s+1} + \alpha_{s+2}| + |\lambda_{s+2}| = 0$ *and* $|\beta_s + 2(b_0\alpha_{s+2} + \alpha_{s+1})| + |\lambda_{s+1}| \neq 0$*, then* $\mathfrak{s}_1 = \mathfrak{s} - 1$*.*
- *If* $|\beta_{s+1} + \alpha_{s+2}| + |\lambda_{s+2}| + |\beta_s + 2(b_0\alpha_{s+2} + \alpha_{s+1})| + |\lambda_{s+1}| = 0$*, then* $\mathfrak{s}_1 = \mathfrak{s} - 2$*.*

### 3.1. Linear Spectral Transformation on Laguerre-Hahn Functional

Now, we will deduce some results concerning the Christoffel and Geronimus transformation when the original linear functional **u** is Laguerre-Hahn.

#### 3.1.1. Christoffel Transformation

**Theorem 6.** *Let* $\widetilde{\mathbf{u}} = (x - c)\mathbf{u}$*. If* **u** *is Laguerre-Hahn functional of class* $\mathfrak{s}$ *with* $\mathbf{u}_0 = \widetilde{\mathbf{u}}_0 = 1$ *satisfying* (19)*, then* $\widetilde{\mathbf{u}}$ *is also a Laguerre-Hahn functional satisfying the equation*

$$\widetilde{\phi}(z) S'_{\widetilde{\mathbf{u}}}(z) = \widetilde{A}(z) S^2_{\widetilde{\mathbf{u}}}(z) + \widetilde{B}(z) S_{\widetilde{\mathbf{u}}}(z) + \widetilde{C}(z), \tag{24}$$

*where*

$$\widetilde{\phi}(z) = (z - c)\phi(z),$$
$$\widetilde{A}(z) = A(z),$$
$$\widetilde{B}(z) = \phi(z) + 2A(z) + (z - c)B(z),$$
$$\widetilde{C}(z) = \phi(z) + 2A(z) + (z - c)B(z) + (z - c)^2 C(z).$$

*Moreover, the class of* $\widetilde{\mathbf{u}}$*, denoted by* $\widetilde{\mathfrak{s}}$*, satisfies* $\mathfrak{s} - 2 \leq \widetilde{\mathfrak{s}} \leq \mathfrak{s} + 1$*.*

**Proof.** (24) is a direct consequence from the fact that you can write, by using (6), $S_{\mathbf{u}}(z)$ in terms of $S_{\widetilde{\mathbf{u}}}(z)$ and then you replace it in (19). Moreover, the linear functional $\widetilde{\mathbf{u}}$ satisfies the distributional equation

$$\mathcal{D}(\widetilde{\phi}(x)\widetilde{\mathbf{u}}) + \widetilde{\psi}(x)\widetilde{\mathbf{u}} - \widetilde{A}(x)(x^{-1}\widetilde{\mathbf{u}}^2) = 0,$$

with $\widetilde{\psi}(x) = (x - c)\psi(x) - 2[\phi(x) + A(x)]$ and $\psi(x) = -[\phi'(x) + B(x)]$. Thus, if $\widetilde{\mathfrak{s}}$ is the class of $\widetilde{\mathbf{u}}$, it follows from above that

$$\deg(\widetilde{\phi}) =: \widetilde{r} \leq \mathfrak{s} + 3, \qquad\qquad \deg(\widetilde{\psi}) =: \widetilde{t} \leq \mathfrak{s} + 2,$$
$$\widetilde{d} = \max\{\widetilde{r}, \widetilde{m}\} \leq \mathfrak{s} + 3, \qquad\qquad \widetilde{\mathfrak{s}} \leq \max\{\widetilde{t} - 1, \widetilde{d} - 2\} \leq \mathfrak{s} + 1,$$

where $\deg \widetilde{B}(x) = \widetilde{m}$.

On the other hand, since $\widetilde{\mathbf{u}}$ is a Laguerre-Hahn functional of class $\widetilde{\mathfrak{s}}$, then there exist polynomials $\overline{\phi}(x)$, $\overline{\psi}(x)$ and $\overline{A}(x)$ such that

$$\mathcal{D}(\overline{\phi}(x)\widetilde{\mathbf{u}}) + \overline{\psi}(x)\widetilde{\mathbf{u}} + \overline{A}(x)(x^{-1}\widetilde{\mathbf{u}}^2) = 0, \tag{25}$$

and $\widetilde{\mathfrak{s}} := \max\{\deg \overline{\psi}(x) - 1, \max\{\deg \overline{\phi}(x), \deg \overline{A}(x)\} - 2\}$. Using (6) again, and taking into account (25), we have that **u** also satisfies the distributional equation

$$\mathcal{D}((x - c)\overline{\phi}(x)\mathbf{u}) + (x - c)[\overline{\psi}(x) + 2\overline{A}(x)]\mathbf{u} + (x - c)^2\overline{A}(x)(x^{-1}\mathbf{u}^2) = 0.$$

With this in mind,

$$\deg(\phi) = r \leq \widetilde{\mathfrak{s}} + 3, \qquad\qquad \deg(\psi) = t \leq \widetilde{\mathfrak{s}} + 2,$$

$$d = \max\{r, m\} \leq \widetilde{\mathfrak{s}} + 4, \qquad \mathfrak{s} \leq \max\{t - 1, d - 2\} \leq \widetilde{\mathfrak{s}} + 2.$$

As a conclusion, $\mathfrak{s} - 2 \leq \widetilde{\mathfrak{s}} \leq \mathfrak{s} + 1$. $\quad \square$

The above gives bounds for the class of $\widetilde{\mathbf{u}}$. However, the following result show that the class only depends on the value $c$ and never take the value $\mathfrak{s} - 2$.

**Theorem 7 (Reference [10]).** *Let $\widetilde{\mathbf{u}} = (x - c)\mathbf{u}$ be a linear functional such that $\mathbf{u}_0 = \widetilde{\mathbf{u}}_0 = 1$ and where $\mathbf{u}$ is of class $\mathfrak{s}$. If $\widetilde{\mathfrak{s}}$ is the class of $\widetilde{\mathbf{u}}$, then*

- $\widetilde{\mathfrak{s}} = \mathfrak{s} + 1$, *if $\phi(c) \neq 0$ and $A(c) \neq 0$.*
- $\widetilde{\mathfrak{s}} = \mathfrak{s}$, *if $\phi(c) = A(c) = 0$, $\psi(c) \neq 0$ and $A'(c) \neq 0$.*
- $\widetilde{\mathfrak{s}} = \mathfrak{s} - 1$, *if $\phi(c) = A(c) = \psi(c) = A'(c) = 0$.*

### 3.1.2. Geronimus Transformation

**Theorem 8.** *Let $\widehat{\mathbf{u}}$ be the linear functional defined by $(x - c)\widehat{\mathbf{u}} = \mathbf{u}$. Assume that $\mathbf{u}_0 = \widehat{\mathbf{u}}_0 = 1$. If $\mathbf{u}$ is a Laguerre-Hahn functional of class $\mathfrak{s}$ satisfying (19), then so is $\widehat{\mathbf{u}}$. In this case*

$$\widehat{\phi}(z) S'_{\widehat{\mathbf{u}}}(z) = \widehat{A}(z) S^2_{\widehat{\mathbf{u}}}(z) + \widehat{B}(z) S_{\widehat{\mathbf{u}}}(z) + \widehat{C}(z), \tag{26}$$

*where*

$$\widehat{\phi}(z) = (z - c)\phi(z), \tag{27}$$
$$\widehat{A}(z) = (z - c)^2 A(z), \tag{28}$$
$$\widehat{B}(z) = -[\phi(z) + 2(z - c)A(z)] + (z - c)B(z),$$
$$\widehat{C}(z) = A(z) - B(z) + C(z).$$

*The class $\widehat{\mathfrak{s}}$ of $\widehat{\mathbf{u}}$ depends only on the zero $x = c$. Moreover, $\mathfrak{s} - 1 \leq \widehat{\mathfrak{s}} \leq \mathfrak{s} + 2$.*

**Proof.** Let $a$ be a zero of $\phi(x)$, and then

$$\widehat{\phi}'(a) + \widehat{\psi}(a) = (a - c)(\phi'(a) + \psi(a) + 2A(a)), \quad \widehat{A}(a) = (a - c)^2 A(a) \tag{29}$$

and

$$\langle \mathbf{u}, \theta_a \widehat{\psi}(x) + \theta_a^2 \widehat{\phi}(x) - (\mathbf{u} * \theta_0 \theta_a \widehat{A})(x) \rangle$$
$$= \langle \mathbf{u}, \theta_a \psi(x) + \theta_a^2 \phi(x) - (\mathbf{u} * \theta_0 \theta_a A)(x) \rangle + \psi(a) + \phi'(a) + A(a), \tag{30}$$

*where*

$$\widehat{\psi}(x) = (x - c)[\psi(x) + 2A(x)]. \tag{31}$$

Observe that, from (27), (28), and (31), we get (29) in a straightforward way. Now, to find (30), let us notice that

$$\theta_a \widehat{\psi}(x) = (x - c)\theta_a \psi(x) + 2\widehat{\mathbf{u}}_0 \theta_a A(x) + \psi(a) + 2\widehat{\mathbf{u}}_0 A(a), \tag{32}$$
$$\theta_a^2 \widehat{\phi}(x) = (x - c)\theta_a^2 \phi(x) + \phi'(a). \tag{33}$$

On the other hand, using (28) and Proposition 1 (ii), we get

$$(\theta_0 \theta_a A)(x) = \theta_0((x - c)^2 \theta_a A(x)) + A(a)$$
$$= (x - c)^2 (\theta_0 \theta_a A)(x) + (x - 2c)(\theta_a A)(0) + A(a).$$

Thus, from the above and Proposition 1 (iii) and (iv),

$$\left\langle \widehat{\mathbf{u}}^2, (x - c)^2 \theta_0 \theta_a A(x) \right\rangle = \left\langle \mathbf{u}^2, (\theta_0 \theta_a A)(x) \right\rangle + 2\widehat{\mathbf{u}}_0 \langle x\mathbf{u}, \theta_0 \theta_a A(x) \rangle,$$

$$\left\langle \widehat{\mathbf{u}}^2, (x - 2c)(\theta_0 A)(0) \right\rangle = \widehat{\mathbf{u}}_0((2 - c) + x\widehat{\mathbf{u}}_0)(\theta_a A)(0),$$

$$\left\langle \widehat{\mathbf{u}}^2, A(a) \right\rangle = \widehat{\mathbf{u}}_0 A(a).$$

Taking into account (32), (33), and the fact that $(\theta_a A)(x) - (\theta_a A)(0) = x\theta_0\theta_a A(x)$, we get (30). Using the above, we obtain that, for any zero $a \neq c$ of $\phi(x)$,

$$|\widehat{\psi}(a) + \widehat{\phi}'(a)| + |\widehat{A}(a)| + \left|\langle \mathbf{u}, \theta_a\widehat{\psi}(x) + \theta_a^2\widehat{\phi}(x) - (\mathbf{u} * \theta_0[\theta_a\widehat{A}(x)])\rangle\right| \neq 0.$$

The proof of the second part is essentially the same as the one given in Theorem 6, and, as a consequence, we do not deal with. However, we point out that $\widehat{\mathbf{u}}$ satisfies the distributional equation

$$\mathcal{D}((x - c)\phi(x)\widehat{\mathbf{u}}) + (x - c)[\psi(x) + 2\widehat{\mathbf{u}}_0 A(x)]\widehat{\mathbf{u}} + (x - c)^2 A(x)(x^{-1}\widehat{\mathbf{u}}^2) = 0.$$

$\square$

**Proposition 9.** *Let $(x - c)\widehat{\mathbf{u}} = \mathbf{u}$, and let $\mathfrak{s}$ and $\widehat{\mathfrak{s}}$ be the class of $\mathbf{u}$ and $\widehat{\mathbf{u}}$, respectively. Let us define*

$$g = \begin{cases} \mathfrak{s} + 2, & \deg A = \mathfrak{s} + 2, \\ \mathfrak{s} + 1, & \deg A < \mathfrak{s} + 2. \end{cases}$$

*Then,*

$$\phi(c) \neq 0 \Rightarrow \widehat{\mathfrak{s}} = g$$

$$\phi(c) = 0 \Rightarrow \begin{cases} A(c) - B(c) + C(c) \neq 0 \Rightarrow \widehat{\mathfrak{s}} = g, \\ A(c) - B(c) + C(c) = 0 \Rightarrow [1], \end{cases}$$

$$[1] \Rightarrow \begin{cases} -[\phi'(a) + 2A(c)] + B(c) \neq 0 \Rightarrow \widehat{\mathfrak{s}} = g - 1, \\ -[\phi'(a) + 2A(c)] + B(c) = 0 \Rightarrow \begin{cases} A'(c) - B'(c) + C'(c) \neq 0 \Rightarrow \widehat{\mathfrak{s}} = g - 1, \\ A'(c) - B'(c) + C'(c) = 0 \Rightarrow [2], \end{cases} \end{cases}$$

$$[2] \Rightarrow \begin{cases} \phi'(c) \neq 0 \Rightarrow \widehat{\mathfrak{s}} = g - 2, \\ \phi'(c) = 0 \Rightarrow \begin{cases} A(c) \neq 0 \Rightarrow \widehat{\mathfrak{s}} = g - 2, \\ A(c) = 0 \text{ is not possible.} \end{cases} \end{cases}$$

**Proof.** Notice that $\deg \widehat{\phi}(x) \leq \mathfrak{s} + 3$, $\deg \widehat{A}(x) \leq \mathfrak{s} + 4$, and $\deg \widehat{\psi}(x) \leq \max\{\mathfrak{s} + 1, \mathfrak{s} + 2\}$, where $\widehat{\phi}(x) = (x - c)\phi(x)$, $\widehat{A}(x) = (x - c)^2 A(x)$, and $\widehat{\psi}(x) = (x - c)(\psi(x) + 2A(x))$. Since

$$\widehat{\mathfrak{s}} = \max\left\{\deg \widehat{\psi}(x) - 1, \ \max\{\deg \widehat{\phi}(x), \deg \widehat{A}(x)\} - 2\right\},$$

if $\phi(c) \neq 0$, then there are two possibilities. If $\deg(A) = \mathfrak{s} + 2$, then $\widehat{\mathfrak{s}} = \mathfrak{s} + 2$. If $\deg(A) < \mathfrak{s} + 2$, then $\widehat{\mathfrak{s}} = \mathfrak{s} + 1$. Now, if $\phi(c) = 0$ and $A(c) - B(c) + C(c) = 0$, then we can divide both sides in (26) by $(z - c)$

$$\phi(z)S_{\widehat{\mathbf{u}}}'(z)$$
$$= (z - c)A(z)S_{\widehat{\mathbf{u}}}^2(z) + \left(-\left[\frac{\phi(z)}{(z - c)} + 2A(z)\right] + B(z)\right)S_{\widehat{\mathbf{u}}}(z) + \frac{A(z) - B(z) + C(z)}{(z - c)}.$$

If $-[\phi'(a) + 2A(c)] + B(c) \neq 0$, then $\widehat{\mathfrak{s}} = g - 1$. On the other hand, if

$$-[\phi'(a) + 2A(c)] + B(c) = 0 \tag{34}$$

and $A'(c) - B'(c) + C'(c) = 0$, then, in (26), we can divide both sides by $(z - c)^2$, and, as a consequence,

$$\frac{\phi(z)}{z - c} S'_{\widehat{\mathbf{u}}}(z) = A(z) S^2_{\widehat{\mathbf{u}}}(z) + \left( -\frac{\phi(z)}{(z - c)^2} + \frac{[2A(z) + B(z)]}{(z - c)} \right) S_{\widehat{\mathbf{u}}}(z) + \frac{A(z) - B(z) + C(z)}{(z - c)^2}.$$

If $\phi'(c) \neq 0$, then $\widehat{\mathfrak{s}} = g - 2$. Finally, if $\phi'(c) = 0$, then $A(c) \neq 0$ since, otherwise, $B(c) = 0$ by (34). This yields the Equation (21) is reducible contradicting the class of $\mathbf{u}$. □

## 4. Examples

**Example 1.** *Let $(P_n^{(\alpha,\beta)}(x))_{n \geq 0}$ be the monic Jacobi polynomials of parameters $(\alpha, \beta)$. For $\alpha, \beta > -1$, these polynomials are orthogonal with respect to the positive definite linear functional $\mathbf{u}^{(\alpha,\beta)}$ defined by*

$$\left\langle \mathbf{u}^{(\alpha,\beta)}, p(x) \right\rangle = \int_{-1}^{1} p(x)(1 - x)^\alpha (1 + x)^\beta dx, \quad p(x) \in \mathbb{P}.$$

*The explicit expression for these polynomials is*

$$P_n^{(\alpha,\beta)}(x) = \frac{1}{S_n(\alpha, \beta)} \sum_{k=0}^{n} \binom{n + \alpha}{n - k} \binom{n + \beta}{k} (x - 1)^k (x + 1)^{n-k}, \; n \geq 0,$$

*where*

$$S_n(\alpha, \beta) = \binom{2n + \alpha + \beta}{n}.$$

*Here, $\binom{r}{k} = \frac{\Gamma(r+1)}{\Gamma(k+1)\Gamma(r-k)}$ and $\Gamma(z) = \int_0^\infty t^{z-1} e^{-t} dt$ is the Gamma function. The orthogonality relation reads*

$$\left\langle \mathbf{u}^{(\alpha,\beta)}, P_n^{(\alpha,\beta)}(x) P_m^{(\alpha,\beta)}(x) \right\rangle = 2^{2n+\alpha+\beta+1} \frac{\Gamma(n + \alpha + 1)\Gamma(n + \beta + 1)\Gamma(n + \alpha + \beta + 1)n!}{(2n + \alpha + \beta + 1)(\Gamma(2n + \alpha + \beta + 1))^2} \delta_{n,m}.$$

*The monic Jacobi polynomials satisfy the three term recurrence relation (see Reference [2])*

$$xP_n^{(\alpha,\beta)}(x) = P_{n+1}^{(\alpha,\beta)}(x) + b_n P_n^{(\alpha,\beta)}(x) + a_n P_{n-1}^{(\alpha,\beta)}(x), \quad n \geq 0,$$
$$P_0^{(\alpha,\beta)}(x) = 1 \; P_{-1}^{(\alpha,\beta)}(x) = 0,$$

*where*

$$b_n = \frac{\beta^2 - \alpha^2}{(2n + 2 + \alpha + \beta)(2n + \alpha + \beta)}, \quad n \geq 0,$$

$$a_n = \frac{4(n + \beta)(n + \alpha + \beta)(n + \alpha)n}{(2n - 1 + \alpha + \beta)(2n + \alpha + \beta)^2(2n + \alpha + \beta + 1)}, \quad n \geq 1,$$

*except that when $\alpha = -\beta$, $b_0 = \beta$ and $b_n = 0$, $n \geq 1$. In particular, if $\alpha = \beta = -1/2$, the polynomials $P_n^{(-\frac{1}{2}, -\frac{1}{2})} =: \widehat{T}_n(x)$ are said to be the monic Chebyshev polynomials of the first kind. If $\alpha = \beta = 1/2$, the polynomials $P_n^{(\frac{1}{2}, \frac{1}{2})} =: \widehat{U}_n(x)$ are said to be the monic Chebyshev polynomials of the second kind. Let $\mathbf{u} =: \mathbf{u}^{(-\frac{1}{2}, -\frac{1}{2})}$, and then it is clear that, for $c = -1$, $\widetilde{\mathbf{u}} = (x + 1)\mathbf{u} = \mathbf{u}^{(-\frac{1}{2}, \frac{1}{2})}$.*

Taking into account that $\widehat{T}_n^{(1)}(x) = \widehat{U}_n(x)$ for all $n \geq 0$ and $\widetilde{\mathbf{u}^{(1)}} = (x+1)\mathbf{u}^{(1)} = \mathbf{u}^{(\frac{1}{2},\frac{3}{2})}$ (see Figure 3), then, from Theorem 3, we have the functional relation

$$\mathbf{u}^{(\frac{1}{2},\frac{3}{2})} = (x+1)^2 \left( \left( \mathbf{u}^{(-\frac{1}{2},\frac{1}{2})} \right)^{-1} - \frac{1}{\pi}\delta_0' \right)^{-1}.$$

Notice also $\mathbf{u}^{(\frac{1}{2},\frac{3}{2})} = (x+1)^2 \mathbf{u}^{(\frac{1}{2},-\frac{1}{2})}$; therefore,

$$\mathbf{u}^{(\frac{1}{2},-\frac{1}{2})} = \left( \left( \mathbf{u}^{(-\frac{1}{2},\frac{1}{2})} \right)^{-1} - \frac{1}{\pi}\delta_0' \right)^{-1} + c_1\delta_{-1} + c_2\delta_{-1}'.$$

Using (3) with $a = -1$ and the fact that $\int_{-1}^1 x(1-x)^{1/2}(1+x)^{-1/2}dx = -\pi/2$, we get that $c_1 = c_2 = 0$. Thus, we conclude that

$$P_n^{(\frac{1}{2},-\frac{1}{2})}(x) = P_n^{(-\frac{1}{2},\frac{1}{2})}(x) - \left( P_{n-1}^{(-\frac{1}{2},\frac{1}{2})} \right)^{(1)}(x),$$

as well as

$$(x+1)^2 P_n^{(\frac{1}{2},\frac{3}{2})}(x) =$$

$$\frac{1}{2(n+1)^3} \det \begin{pmatrix} P_{n+2}^{(\frac{1}{2},-\frac{1}{2})}(x) & P_{n+1}^{(\frac{1}{2},-\frac{1}{2})}(x) & P_n^{(\frac{1}{2},-\frac{1}{2})}(x) \\ 1/4 & -1/2 & 1 \\ 1/4(n+2)^2(n+3)^2 & -1/2(n+1)^2(n+2)^2 & n^2(n+1)^2 \end{pmatrix}.$$

**Figure 3.** Perturbation of Jacobi polynomials by a canonical Christoffel transformation with $(x+1)$.

**Example 2.** *Let* $(L_n^{\alpha+1})_{n \geq 0}$ *be the monic Laguerre polynomials of parameter* $\alpha + 1$ *with* $\alpha > -1$, *which are orthogonal with respect to the positive definite linear functional* $\mathbf{v}$

$$\langle \mathbf{v}, p(x) \rangle = \frac{1}{\Gamma(\alpha+2)} \int_0^\infty p(x)x^{\alpha+1}e^{-x}dx, \quad p(x) \in \mathbb{P}.$$

*The following properties are very well known in the literature (Reference [2]):*

*(i)* *Recurrence relation.*

$$xL_n^{\alpha+1}(x) = L_{n+1}^{\alpha+1}(x) + (2n+\alpha+2)L_n^{\alpha+1}(x) + n(n+\alpha+1)L_{n-1}^{\alpha+1}(x), \quad n \geq 0,$$
$$L_0^{\alpha+1}(x) = 1, \qquad L_{-1}^{\alpha+1}(x) = 0.$$

*(ii)* *Explicit formula as an hypergeometric function:*

$$L_n^\alpha(x) = \frac{n!}{(-1)^n} \sum_{k=0}^n \frac{(\alpha)_{n+1}}{(n-k)!(\alpha)_{k+1}} \frac{(-x)^k}{k!},$$

*where* $(a)_n := (a)(a+1)\cdots(a+n-1)$, $(a)_0 = 1$ *is the Pochhammer symbol.*

*(iii)* $\left( \dfrac{d^i}{x^i} L_n^{\alpha+1} \right)(0) = (-1)^{n+i}\dfrac{n!}{(n-i)!}(\alpha+i+2)_{n-i}.$

*(iv)*     $\left\langle \mathbf{v}, L_n^{\alpha+1}(x)L_m^{\alpha+1}(x) \right\rangle = n!(\alpha+2)_n \delta_{n,m}$.

*The linear functional $\mathbf{v}$ satisfies the distributional equation [25] $D(x\mathbf{v}) + (x - \alpha - 2)\mathbf{v} = 0$, and, as a consequence, it is a Laguerre-Hahn functional of class $\mathfrak{s}(\mathbf{v}) = 0$. Its Stieltjes function satisfies the first order linear differential equation*

$$z S'_{\mathbf{v}}(z) = (-z + \alpha + 1)S_{\mathbf{v}}(z) + 1. \tag{35}$$

*In Reference [36], the authors studied the first kind associated Laguerre polynomials which are denoted by $(L_n^{\alpha+1}(x,1))_{n\geq 0}$. In particular, it was proven that these polynomials are orthogonal with respect to the positive definite functional $\mathbf{v}^{(1)}$ defined by*

$$\left\langle \mathbf{v}^{(1)}, p(x) \right\rangle = \frac{1}{\Gamma(\alpha+3)} \int_0^\infty p(x) \frac{x^{\alpha+1}e^{-x}}{\left|\Psi(1,-\alpha, xe^{-\pi i})\right|^2} dx,$$

*where*

$$\Psi(c,a,x) = \frac{1}{\Gamma(c)} \int_0^{\infty e^{(3\pi/4)i}} t^{c-1}(1+t)^{a-c-1}e^{-xt}dt,$$

$\mathrm{Re}(c) > 0, \; -\pi/2 < 3\pi/4 + \arg x < \pi/2$.

*The associated monic orthogonal polynomials of the first kind $(L_n^{\alpha+1}(x,1))_{n\geq 0}$ satisfy the following properties:*

*(i)*     *Explicit formula.*

$$L_n^{\alpha+1}(x,1) = (-1)^n(n+1)(\alpha+3)_n \times$$
$$\sum_{k=0}^n \frac{(-n)_k x^k}{(k+1)!(\alpha+3)_k} \times {}_3F_2\left( \begin{array}{c} k-n,\, 1,\, \alpha+2 \\ \alpha+k+3,\, k+2 \end{array}; 1 \right).$$

*(ii)*     $L_n^{\alpha+1}(0,1) = \dfrac{(-1)^n}{\alpha+1}\left[(\alpha+2)_{n+1} - (n+1)!\right]$.

*(iii)*     $\left\langle \mathbf{v}^{(1)}, L_n^{\alpha+1}(x,1)L_m^{\alpha+1}(x,1) \right\rangle = (n+1)!(\alpha+3)_n \delta_{n,m}$.

*Using Theorem 5 and (35), we get that the functional $\mathbf{v}^{(1)}$ is Laguerre-Hahn of class zero satisfying the distributional equation*

$$D(x\mathbf{v}^{(1)}) + (x - \alpha - 4)\mathbf{v}^{(1)} - (\alpha+2)\left(x^{-1}\left[\mathbf{v}^{(1)}\right]^2\right) = 0.$$

*From Proposition 7, we have the structure relation*

$$x\frac{d}{dx}L_{n+1}^{\alpha+1}(x,1) + (\alpha+2)L_n^{\alpha+1}(x,2) = \sum_{k=n}^{n+1}\lambda_{n,k}L_k^{\alpha+1}(x,1); \quad n \geq 1.$$

*Using (2) and comparing the coefficients on both sides, we get*

$$\lambda_{n,n+1} = (n+1), \quad \lambda_{n,n} = (n+\alpha+3)(n+2),$$

*as well as the relation*

$$x\frac{d}{dx}L_{n+1}^{\alpha+1}(x,1) - L_{n+2}^{\alpha+1}(x) = (-x+\alpha+n+3)L_{n+1}^{\alpha+1}(x,1) +$$
$$(n+\alpha+3)(n+2)L_n^{\alpha+1}(x,1), \quad n \geq 1.$$

*Besides, from (22), its Stieltjes function satisfies the differential equation*

$$z S'_{\mathbf{v}^{(1)}}(z) = (\alpha+2)S^2_{\mathbf{v}^{(1)}}(z) + (-z+\alpha+3)S_{\mathbf{v}^{(1)}}(z) + 1.$$

Next, let $\widehat{\mathbf{v}}$ be the linear functional defined by the Geronimus transformation $x\widehat{\mathbf{v}} = \mathbf{v}$ with $\widehat{\mathbf{v}}_0 = \dfrac{1}{(\alpha+1)}$. Then, from Section 2.2 with $c = 0$, we get

$$\langle \widehat{\mathbf{v}}, p(x) \rangle = \frac{1}{\Gamma(\alpha+2)} \int_0^\infty (p(x) - p(0)) x^\alpha e^{-x}\, dx + \frac{p(0)}{\Gamma(\alpha+2)} \int_0^\infty x^\alpha e^{-x}\, dx$$

$$= \frac{1}{\Gamma(\alpha+2)} \int_0^\infty p(x) x^\alpha e^{-x} dx, \quad p(x) \in \mathbb{P}.$$

If $J_{\alpha+1}$ is the monic Jacobi matrix associated with $\mathbf{v}$, then $J_{\alpha+1}$ has UL factorization as in (13) with $\beta_n = \alpha + n + 1$, $n \geq 0$, $\ell_n = n$, $n \geq 1$. Therefore,

$$J_{\alpha+1} = UL \longmapsto \widehat{J}_{\alpha+1} =: LU = J_\alpha.$$

Notice that $\widehat{L_n^{\alpha+1}}(x) = L_n^\alpha(x)$ for all $n \in \mathbb{N}$. Now, let $\widehat{\mathbf{v}^{(1)}}$ be the Geronimus transformation of $\mathbf{v}^{(1)}$ obtained from (15), and then

$$\left\langle \widehat{\mathbf{v}^{(1)}}, p(x) \right\rangle = \frac{1}{\Gamma(\alpha+3)} \int_0^\infty (p(x) - p(0)) \frac{x^\alpha e^{-x}}{\left| \Psi(1, -\alpha, xe^{-\pi i}) \right|^2} dx + \frac{p(0)}{(\alpha+2)}.$$

Let $(P_n(x, 1))_{n \geq 0}$ be the SMOP with respect to $\widehat{\mathbf{v}^{(1)}}$. Then, from Corollary 2, we get the three term recurrence relation

$$xP_n(x, 1) = P_{n+1}(x, 1)(x) + (2n + \alpha + 1)P_n(x, 1) + (n+1)(n + \alpha + 1)P_{n-1}(x, 1), \quad n \geq 1,$$
$$P_0(x, 1) = 1 \quad P_1(x, 1) = x - (\alpha + 2),$$

as well as the following connection formula:

$$P_n(x, 1) = \frac{1}{\alpha+1} \left( xL_n^\alpha(x, 1) - L_{n+1}^\alpha(x) \right) \quad n \geq 2.$$

Using Theorem 8, we get that the linear functional $\mathbf{w} := (\alpha + 2)\widehat{\mathbf{v}^{(1)}}$ is a Laguerre-Hahn functional, and

$$z^2 S'_{\mathbf{w}}(z) = (\alpha+2)z^2 S^2_{\mathbf{w}}(z) - z(z + \alpha + 2)S_{\mathbf{w}}(z) + z.$$

Note that its equation is reducible to

$$zS'_{\mathbf{w}}(z) = (\alpha+2)zS^2_{\mathbf{w}}(z) - (z + \alpha + 2)S_{\mathbf{w}}(z) + 1.$$

Taking into account that the polynomials are coprime, then the class of $\mathbf{w}$ is also zero. Since $\widehat{\psi}_1(x) = x + (\alpha + 1)$, then $\mathbf{w}$ satisfies the distributional equation

$$D(x\mathbf{w}) + (x + \alpha + 1)\mathbf{w} - (\alpha + 2)x\left( x^{-1}[\mathbf{w}]^2 \right) = 0.$$

## 5. Concluding Remarks

In this contribution, we have focused our attention on linear and rational spectral transformations of linear functionals and the corresponding action both on the Jacobi matrices and Stieltjes functions associated with them. In particular, in such a framework, we have solved the following intertwining problem. *Given a linear spectral transformation $T_c$, we dealt with the analysis of the transformation $R$ such that $T_c\, T^{(1)} = R\, T_c$, where $T^{(1)}$ denotes the associated transformation of the first kind.* We have applied this fact to the analysis of corresponding polynomials. The behavior of the Laguerre-Hahn linear functionals when one deals with the above transformations has been analyzed, and the class of the resulting linear functional has been discussed. Finally, some illustrative examples concerning Jacobi and Laguerre polynomials have been presented.

**Author Contributions:** Conceptualization, J.C.G.-A. and F.M.; methodology, J.C.G.-A. and F.M.; formal analysis, J.C.G.-A. and F.M.; investigation, J.C.G.-A. and F.M.; resources, J.C.G.-A. and F.M.; data creation, J.C.G.-A. and F.M.; writing original draft preparation, J.C.G.-A. and F.M.; writing review and editing, J.C.G.-A. and F.M.; visualization, J.C.G.-A. and F.M.; supervision, J.C.G.-A. and F.M.; project administration, F.M.; funding acquisition, F.M. All authors have read and agreed to the published version of the manuscript.

**Funding:** The work of the second author (FM) has been supported by Agencia Estatal de Investigación (AEI) of Spain and Fondo Europeo de Desarrollo Regional (FEDER), grant PGC2018-096504-B-C33.

**Institutional Review Board Statement:** Not applicable.

**Informed Consent Statement:** Not applicable.

**Data Availability Statement:** Not applicable.

**Acknowledgments:** We would like to thank the referees for their helpful remarks and suggestions which have contributed to improve the presentation of the manuscript.

**Conflicts of Interest:** The authors declare no conflict of interest.

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
