# Peer review of "Spectral Transformations and Associated Linear Functionals of the First Kind"

_axioms, doi:10.3390/axioms10020107_

Round 1

Reviewer 1 Report

The associated transforms and Darboux transforms are two important techniques to perturb sequences of orthogonal polynomials. This paper and the authors preprint [16] are two twin studies dedicated to the relations between the associated transforms of Darboux transforms and vice versa. (The aim is to complete the commutative diagrams in Fig. 1 and Fig. 2.)  This an interesting and important question within the study of orthogonal polynomials inspired by the relevance of these transformations for applications.

Author Response

The referee #1 has not required any change

Reviewer 2 Report

In the manuscript Spectral transformations and associated linear functionals of the first kind, the authors attempt the exploration of the relations of the Christoffel transformed quasi-definite linear functional in the linear space of polynomials with complex coefficients and their associated orthogonal polynomials of the first kind.

            The manuscript is written in rather clear language, but reading it for a non-specialist in this very specific argument is a painful experience. The work begins with a series of mathematical definitions, propositions and theorems, which have some citations with arbitrary numbering. There is no explanation as to why this argument is addressed, what has been done in the field so far etc. Nothing can be traced in the introductory part, but for the evidence that the authors suppose a reader is well acquainted with the problematic touched and knows all the details. Though of such readers, only the authors are … Furthermore, the introduction is crowned by the following phrase: "Next, we will point out the goals of our contribution. In [16] we study the following problem: Problem1. ", and them a schematic diagram appears. If the above statement relates to [16], what does it have to do with the "goals of the current contribution", unless this work itself is [16]?

            The rest of the work consists of some propositions, theorems etc in following each other.

            The paper falls short in terms of a scientific work for a journal. There is no introduction to the subject, no line of research is shown, no background is explained, neither the significance of the contribution, nor the novelty is underlined. There is nothing in this assembly of propositions and corollaries, which allows the comparison with contributions of others in this field and allows judging on the importance of the problem itself. This is not to mention the complete absence of the discussion, conclusions and results, presented in any decent way.

            We propose that the authors either publish this in the present form as a part of a monograph, or subject it to a COMPLETE revision, addressing the above issues, and submit it to a specialized journal after that. 

Author Response

Enclosed please find the author's reply of the report by referee #2

Reviewer 3 Report

attached file

Author Response

Enclosed please find the author's reply to the report by referee #3

Round 2

Reviewer 2 Report

This Referee appreciates the changes that the authors have introduced in the paper, following the previous review round. One comment regards the fact that in the conclusions use of past tenses is recommended due to the fact that the conclusions describe the completed work. Thus, we advise that the authors change the conclusions accordingly and use consequently one of the past tenses in it. 

One more comment consists in that this Referee wishes the authors explicitly underlined the novelty of their work vs. the contributions of other researchers in the field. To this end, it would be nice to give a brief comparative analysis of the existing achievements in the scientific field studied by the authors, and the respective contribution of the authors on the background of the preceding studies of others. This would allow the reader appreciate the paper even more and demonstrate the significance of the contribution of the authors.  

Author Response

First of all, we thank the valuable comments by the referee. The deep insight in the
subject helped to a substantial improvement of the  presentation of this manuscript.

Q1. The referee wrote :" One comment regards the fact that in the conclusions use of past tenses is recommended due to the fact that the conclusions describe the completed work. Thus, we advise that the authors change the conclusions accordingly and use consequently one of the past tenses in it."

Response. We have changed the conclusions accordingly to the past tenses as suggested by the referee.

Q2. The referee wrote:"  Referee wishes the authors explicitly underlined the novelty of their work vs. the contributions of other researchers in the field. To this end, it would be nice to give a brief comparative analysis of the existing achievements in the scientific field studied by the authors, and the respective contribution of the authors on the background of the preceding studies of others."

Response.  In Section 1  we have included a comment about the new results of our contribution as well s a brief comparative analysis of the scientific achievements in the field.
